# Examination of two different proteasome inhibitors in reactivating mutant human cystathionine β-synthase in mice

**Sapna Gupta, Hyung-Ok Lee, Liqun Wang, Warren D. Kruger** *

Cancer Signaling and Microenvironment Program, Fox Chase Cancer Center, Philadelphia, PA, United States of America

* warren.kruger@fccc.edu

## Abstract

Classic homocystinuria is an inborn error of metabolism caused mainly by missense mutations leading to misfolded and/or unstable human cystathionine β-synthase (CBS) protein, causing the accumulation of excess total homocysteine (tHcy) in tissues. Previously, it has been shown that certain missense containing human CBS proteins can be functionally rescued in mouse models of CBS deficiency by treatment with proteasome inhibitors. The rescue by proteasome inhibitors is thought to work both by inhibiting the degradation of misfolded CBS protein and by inducing the levels of heat-shock chaperone proteins in the liver. Here we examine the effectiveness of two FDA approved protease inhibitors, carfilzomib and bortezomib, on various transgenic mouse models of human CBS deficiency. Our results show that although both drugs are effective in inducing the liver chaperone proteins Hsp70 and Hsp27, and are effective in inhibiting proteasome function, bortezomib was somewhat more robust in restoring the mutant CBS function. Moreover, there was no significant correlation between proteasome inhibition and CBS activity, suggesting that some of bortezomib's effects are via other mechanisms. We also test the use of low-doses of bortezomib and carfilzomib on various mouse models for extended periods of time and find that while low-doses are less toxic, they are also less effective at restoring CBS function. Overall, these results show that while restoration of mutant CBS function is possible with proteasome inhibitors, the exact mechanism is complicated and it will likely be too toxic for long-term patient treatment.

## Introduction

Cystathionine beta synthase (CBS) deficiency (also called classical homocystinuria) is the most common inborn error of sulfur amino acid metabolism, with an estimated prevalence of about 1/100,000 births world-wide [1]. Loss of CBS activity leads to accumulation of total homocysteine (tHcy) in plasma and tissues, resulting in clinical phenotypes including eye lens dislocation, osteoporosis, thrombosis, and learning and behavioral disorders [2]. Current methods to manage CBS deficiency rely on lowering tHcy by restricting dietary intact of the homocysteine precursor methionine and supplementing the diet with B-vitamins and betaine [3].

**Data Availability Statement:** All relevant data are within the paper and its Supporting Information files.

**Funding:** This work was funded by grants from the National Institutes of Health (DK101404 to WK and CA006927 to Fox Chase Cancer Center).The funders had no role in study design, data collection and analysis, decision to publish, or preparation of the manuscript.

**Competing interests:** The authors have declared that no competing interests exist.

The vast majority of mutations associated with CBS deficiency are missense mutations in which a single amino acid in the CBS protein is altered, causing the mutant protein to have difficulty folding into its native active conformation [4]. In most cases this failure to fold properly results in the protein's degradation by the cells proteostasis network. Proteostasis networks include both the machinery to degrade misfolded proteins such as the ubiquitin/proteasome system, and chaperone proteins that aid in the protein folding process [5]. One way to alter the cellular proteostasis network is through the use of drugs that are proteasome inhibitors. This was first shown experimentally using MG132 and celastrol to partially restore function to human fibroblast cell lines harboring missense mutations in the glucocerebrosidase and β-hexosaminidase A genes from Gaucher and Tay-Sachs patients [6]. The two most commonly used FDA approved proteasome inhibitors used in the clinic are bortezomib (Velcade) and carfilzomib (Kyprolis). Bortezomib, used to treat multiple myeloma and mantle cell lymphoma, is a N-pyrazinylcarbonylated derivative of the dipeptide boronic acid Phe-Leu-B(OH)$_2$ that inhibits the proteasome in a reversible manner [7]. It also has off-target effects on other serine or cysteine proteases that may be responsible for some of its side effects, such as peripheral neuropathy [8]. Carfilzomib is a tetrapeptide epoxyketone which irreversibly binds to the proteasome and has greater specificity and clinically is associated with less peripheral neuropathy [9]. A recent phase III trial directly comparing the two drugs in combination with lenalidomide and dexamethasone for patients with newly diagnosed multiple myeloma, showed that both drugs were similarly effective [10].

In previous work, our lab has shown that short-term treatment of mice with bortezomib could restore function to some, but not all, mutant human CBS proteins expressed in an *in vivo* humanized mouse model. Specifically, we found that bortezomib could enhance function of mice expressing the p.I278T, p.R266K, p.S466L, and p.R336C alleles, but was not effective in restoring function to the p.G307S allele [11–14]. In all cases we found that treatment with bortezomib increased the steady-state levels of mutant CBS proteins and induced several chaperone proteins including HSP70 and HSP26. In the studies described here, we compare the relative effectiveness of carfilzomib and bortezomib on several different mouse models in order to determine the relative effectiveness of the drugs, and to gain some mechanistic insight into the relationship between proteasome inhibition, chaperone induction, and functional rescue of mutant CBS. We also examined whether either drug was effective at doses that lack the toxicity of that typically used to treat cancer patients. Our results show that bortezomib, although reversible and less specific compared to carfilzomib, is more effective in restoring significant enzyme activity to mutant CBS alleles *in vivo*. Moreover, there was not much correlation between proteasome inhibition and CBS enzyme activity regardless of treatment, suggesting the involvement of other factors in restoration of mutant protein function in addition to the proteasome. Overall, our data suggests that there is a complex interaction between proteasome inhibitors, chaperone induction, and refolding of mutant CBS.

## Materials and methods

### Mouse models

The *Tg-S466L Cbs*$^{-/-}$, *Tg-R266K Cbs*$^{-/-}$, *Tg-R336C Cbs*$^{-/-}$ and *Tg-I278T Cbs*$^{-/-}$ mouse models have been previously described [12,13,15,16]. For simplification the *Cbs*$^{-/-}$ genotype has been omitted throughout the text. The *Tg-T191M* model was created similarly to the models described above. In brief, site-directed mutagenesis was used to introduce the T191M (c.572C>T) mutation into plasmid pLW3, which contains the human CBS cDNA downstream of the mouse MT-I promoter flanked by long range control elements. These long range control elements allow robust zinc inducible expression of the transgene regardless of integration site.

The plasmid was then linearized, purified, and injected into mouse pronuclei of day 0.5 C57B6/C3H F2 zygotes, which were then implanted into pseudo-pregnant females. From the implantations, one male offspring was positive for the transgene and was crossed to a $Cbs^{+/-}$ female (C57BL6). $Tg+ Cbs^{+/-}$ offspring from this cross were then intercrossed. Like the $Tg$-$R266K$ and $Tg$-$R336C$ alleles, $Tg$-$T191M$ was not able to rescue the neonatal lethality associated with homozygosity for the $Cbs^-$ allele on a C57BL6 background, so we then proceeded to cross animals into a C3H background for three generations. On this background, ~70% of the embryos survived the neonatal period. These survivors were used in the experiments described. Mice were fed the standard rodent chow (Teklad 2018SX) and eight to 10 days before starting the drug treatment mice were put on 25 µM ZnSO4 water to induce transgene expression. Almost all of the mice (92.4%) used in these studies were female, with a median age of 5.6 months. Female mice were used preferentially because previous studies have shown that they have more consistent expression in transgene induction by zinc (11). Details of each animal used for each experiment can be found in S1 File. All experiments utilizing mice were approved by the Fox Chase Cancer Center IACUC.

## Drug studies

Bortezomib (Velcade, Millennium Pharmaceuticals, Cambridge, MA, USA) was obtained from the pharmacy of Fox Chase Cancer Center in reconstituted form at the concentration of 1 mg/ml in 0.9% NaCl. For normal dose studies, Bortezomib was given at 0.49 mg/kg/day using an Alzet micro-osmotic pumps (Model 1007D) subcutaneously implanted on the backs of the mice. Control mice were implanted with pumps containing 0.9% NaCl, the diluent for bortezomib. For the low-dose studies, mice were given a subcutaneous injection of either 0.5mg or 0.8 mg according to the schedule shown in Figs 5 and 6. Mice were euthanized by iso-flurane overdose.

Carfizomib (Kyprolis, Onyx/Amgen, San Francisco, CA, USA) was obtained from the pharmacy of Fox Chase Cancer Center in reconstituted in water. Carfilzomib was given intrave-nously by injecting 150 ul into the retro orbital space. $Tg$-$S466L$ mice were also treated by implanting either an Alzet model 2001D micro-osmotic pump (Fig 1) or a 1007D pump (Fig 7).

## Extract preparation, CBS enzyme assay, and western blotting

Liver tissue lysates were prepared as previously described [11]. For CBS activity assays, extracts were first dialyzed in 10 mM Tris-HCl (pH 8) with 15% glycerol at 4°C using Slide-A-Lyzer Dialy-sis Cassettes (10K MWCO, 0.1–0.5 mL Capacity; Thermo Scientific Catalog# 66383). Assay mix-ture contained 30µg of dialyzed liver extracts, BICINE buffer (20 mM; pH 8.6), 5 mM L-serine, 10 mM DL-homocysteine, 50µM pyridoxal phosphate and 250 µM AdoMet. Cystathionine forma-tion was measured using an ARACUS amino acid analyzer (MembraPure). One unit of activity is defined as nmoles of cystathionine formed per milligram of protein per hour.

For Westerns, protein extract was electrophoresed on either 7% NuPAGE Tris-Acetate or 12% Bis Tris gels (Invitrogen, CA) under denaturing conditions followed by transfer on to a polyvinylidene fluoride (PVDF) membrane (Bio Rad, CA). CBS was detected using our lab's polyclonal anti-human primary antibody (1:10,000) and secondary anti-rabbit antibody (1: 30,000) conjugated with horseradish peroxidase (Amersham Biosciences, UK). For chaperone proteins the following antibodies were purchased from Cell Signaling Technology: HSP90 (#4874); HSP40 (#4868); HSP27 (#2442). Antibody from Stressgen (Hsp70/72, mAb C92F3A-5 #SPA-810) was used to detect Hsp70. Signal was visualized by SuperSignal West Pico Chemi-luminescent kit (Thermo Scientific) and signal was captured using Alpha Innotech image ana-lyzer. Raw data for all Westerns is found in S2 File.

## Proteasome activity measurement

Proteasome activity assay was conducted for the total 20S proteasome which includes latent 20S, 19S-20S (also called 26S) and PA28-20S (also called immunoproteasome). Homogenates (10%) of liver tissues were made in lysis buffer containing 50 mM HEPES (pH 7.5), 10% glycerol, 100 mM NaCl and 0.5 mM DTT. The fluorescent substrate used to detect activity was Suc-Leu-Leu-Val-Tyr-AMC (Cayman Chemical). The assay buffer contained 50 mM HEPES (pH 7.5), 0.5 mM DTT, 0.05% SDS, 250 μM Suc-LLVY-AMC and 70 μg protein. All the assays were done in the absence and presence of irreversible proteasome inhibitor carfilzomib (10μM) to control for non-proteasome associated chemotrypsin-like activity. The Reactions were incubated for 2.5 hour at 37°C. The difference between the two conditions (+/- carfilzomib) was used to determine specific proteasome activity. Quantification was done by using a Standard curve varying the concentration of 7-amino-4-methylcoumarin (AMC) in assay buffer from 0.5 to 250 μM. The production of AMC was monitored using excitation filter of 400 nm and emission filter of 460 nm.

## tHcy measurements

Serum total homocysteine (tHcy) was measured using a Biochrom 30 amino acid analyzer as previously described [15]. In brief, 50μl serum was reduced by 12% Dithiothreitol, followed by precipitation of protein using 10% sulfosalicylic acid. Resulting supernatant was then loaded into the analyzer.

**Blood analysis.** Blood counts were obtained using an Abaxis VetScan 5 Hematology Analyzer.

## Statistics

Data was analyzed using GraphPad Prism 9 software. For data sets involving comparison of two measures, t-tests were used with $p < 0.05$ considered significant. For data sets involving three or more comparisons, one-way ANOVA was used with a Tukey test for comparison of specific means.

## Results

### Carfilzomib rescue of *Tg-S466L Cbs$^{-/-}$* mice

The p.S466L mutation was first described in a CBS deficient patient that presented later in life with thrombosis [17]. *Tg-S466L* mice are homozygous for a null allele of the endogenous mouse *Cbs* gene and express the p.S466L human CBS protein from a zinc inducible metallothionein promoter [18,19]. Zinc-treated *Tg-S466L* mice have an order of magnitude increase in serum tHcy and nearly undetectable levels of CBS protein and activity in liver lysates, suggesting that the mutation affects protein stability. Previously our lab found that treatment of these mice with the experimental oral proteasome inhibitor ONX-0914 results in stabilization of the p.S466L CBS protein in the liver and a significant decreases in plasma homocysteine levels [11].

Due to the robust rescue phenotype of *Tg-S466L*, we performed pilot studies examining the relative effectiveness of two different routes of administration of carfilzomib; intravenous by retro-orbital injection (R.O.) and subcutaneous (S.C.). For the R.O. route the entire bolus of drug (10 mg/kg) was given at time 0, while for the S.C. route the drug (15 mg/kg) was given over a 24-hour period using an osmotic pump. We found that at the end of 24 hours both delivery methods significantly lowered serum tHcy (Fig 1A), increased liver lysate CBS activity levels (Fig 1B) and increased liver CBS protein levels (Fig 1C) compared to untreated control

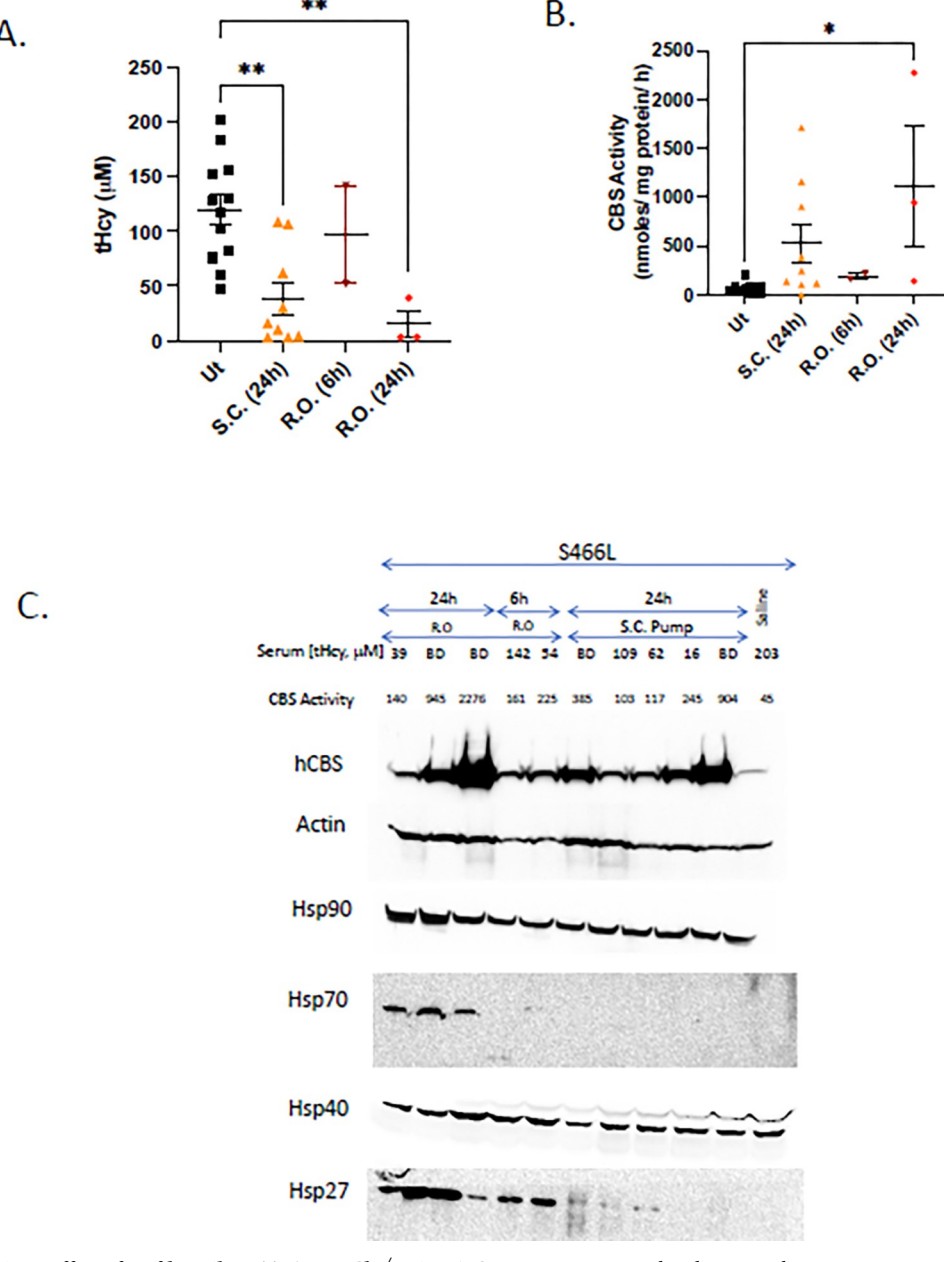

**Fig 1. Effect of carfilzomib on *Tg-S466L Cbs*⁻/⁻ mice.** A. Serum tHcy in treated and untreated mice. Ut = untreated, S.C. = subcutaneous delivery using an osmotic pump (15 mg/kg), R.O. = retro-orbital injection (10 mg/kg). For retro-orbital (R.O.) injected animals two time points were examined. Number of animals and standard error of mean (SEM) is shown. Asterisks indicate significant differences from untreated samples: *p<0.05, **p<0.01. B. Liver CBS activity of treated and untreated mice. C. Western blot analysis of R.O. and S.C. treated animals. Saline-treated control is shown on right. At top of panel, tHcy and liver CBS activity for each mouse is also shown.

animals. We also examined the levels of several heat shock protein chaperones in treated and untreated mice (Fig 1C). We observed robust induction of Hsp70 and Hsp27 in R.O. treated mice after 24 hours, but did not observe induction in S.C. mice. In none of the treated mice did we observe induction of Hsp40 or Hsp90. These findings indicate that treatment with carfilzomib, like ONX-0914, could stabilize and rescue p.S466L *in vivo*.

## Comparison of Carfilzomib and bortezomib rescue of various CBS mutations

We next examined the ability of carfilzomib to rescue four additional mouse models of human CBS deficiency: *Tg-R266K*, *Tg-I278T*, *Tg-R336C*, and *Tg-T191M*. All but the *Tg-T191M* ⁻mice have been previously described [12,13,16]. The *Tg-T191M* mouse expresses a non-pyridoxine responsive mutant CBS protein found at high frequency in patients from in Spanish speaking countries [20] and the creation is described in the Methods section. For each of these alleles, we compared the effectiveness of high dose R.O. delivered carfilzomib with S.C. bortezomib treatment. In each case we measured tHcy, liver lysate CBS activity, liver lysate proteasome activity, and the steady-state protein levels of CBS, Hsp70, Hsp90, and Hsp27.

For all of these alleles, we observed the highest levels of CBS protein in bortezomib-treated mice, but did see lesser increases in carfilzomib treated mice compared to untreated animals (Fig 2A–2D). This was also true with regards to liver lysate CBS activity. In the *Tg-R266K*, *Tg-R336 C*, and *Tg-I278T* strains we found that bortezomib-treatment was much more effective than carfilzomib in restoring CBS activity (Fig 3A). Neither proteasome inhibitor was able to restore significant enzymatic activity to the *Tg-T191M* strain, although treatment with bortezomib did seem to stimulate a very small amount of enzyme activity. In terms of tHcy lowering, we again observed that the bortezomib-treated mice had responded significantly better than the carfilzomib-treated ones (Fig 3B). In fact, only in the *Tg-I278T* strain was tHcy lower than in carfilzomib treated mice compared to untreated control mice and this difference was not statistically significant. Taken together, our results indicate that carfilzomib is not as effective as bortezomib in restoration of function to mutant CBS protein *in vivo*.

## Chaperone induction and proteasome activity

Previously it had been shown that the chaperone proteins Hsp70 and Hsp27 were induced by bortezomib [11,21] in *Tg-I278T* mice. Therefore, we compared the levels of induction in untreated, bortezomib-treated, and carfilzomib-treated mice expressing that various human CBS proteins (Fig 2A–2D). In *Tg-R266K*, *Tg-R336C*, and *Tg-T191M* mice, carfilzomib-treatment showed greater induction of both Hsp70 and Hsp27. In *Tg-I278T* mice, carfilzomib showed greater induction of Hsp70, but bortezomib showed greater induction of Hsp27. In addition, we found that untreated *Tg-I278T* mice had significant levels of Hsp70 and Hsp27 even in the absence of proteasome inhibitor, suggesting that expression of p.I278T CBS itself was causing some level of proteotoxic stress. For several of the mutants and in mice expressing wild type human CBS (hCBS), we also examined two additional chaperones, Hsp90 and Hsp40. We did not observe any evidence that the steady-state levels of these chaperones were affected by either bortezomib and carfilzomib treatment (Fig 1C).

Because of differences in the chemistry and delivery method of the two drugs, and differences in the mutant allele that was being expressed, we decided to compare the effectiveness of the two different treatments in inhibiting proteasome function. Proteasome function was determined by measuring cleavage of a fluorogenic chemotypsin-like substrate in liver lysates in the presence and absence of a specific proteasome inhibitor (Fig 4). Overall, with the exception of the *Tg-R266K* strain, both proteasome inhibitors were equally effective in reducing proteasome activity. For the *Tg-R266K* mice, bortezomib appeared to be somewhat more effective, but the number of mice examined was too small to be statistically significant. To get a better understanding of the relationship between proteasome inhibition and restoration of CBS enzyme activity, for each strain, we plotted proteasome activity vs. liver CBS activity in untreated, carfilzomib-, and bortezomib-treated animals. For each strain we saw a overall negative correlation between CBS activity and residual proteasome activity, but the strength of the

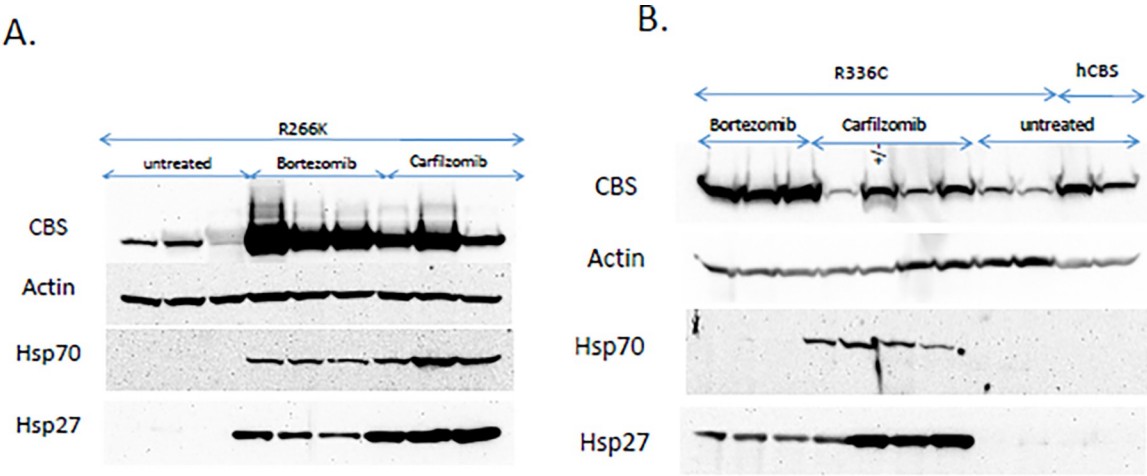

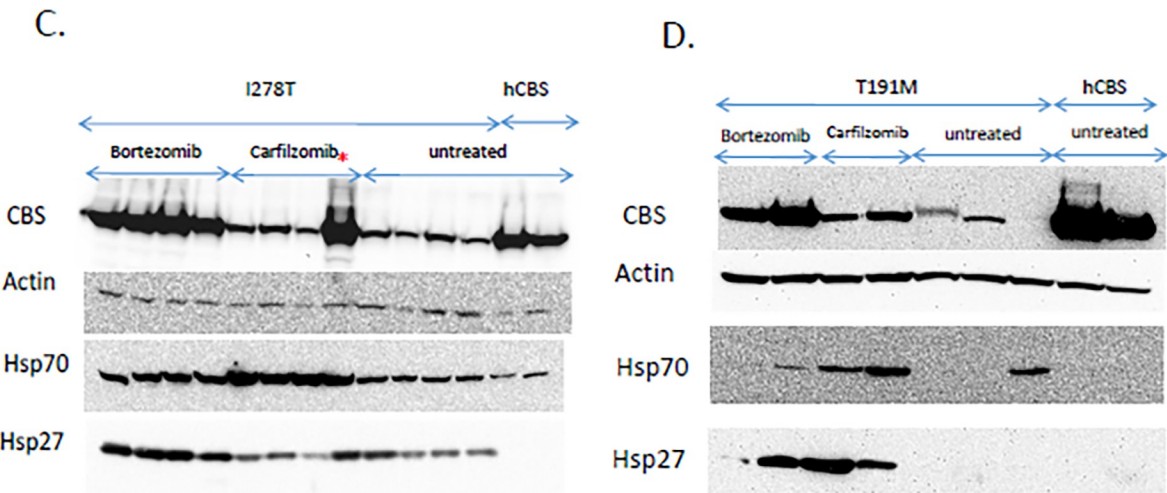

**Fig 2. Western blot analysis of humanized CBS mice treated with either nothing, carfilzomib or bortezomib.** Carfilzomib was administered as a single dose by R.O. injection at 10 mg/kg. Bortezomib was administered via osmotic pump delivering 0.49 mg/kg/day. Carfilzomib mice were euthanized 24 hours after injection, while bortezomib mice were euthanized 36–48 hours after implantation. A. Blot showing liver extracts from *Tg-R266K* mice. B. Liver extracts from *Tg-R336C* mice. C. Liver extracts from *Tg-I278T* mice. Red asterisk denotes highly responsive sample. D. Liver extracts from *Tg-T191M* mice. Lanes labeled hCBS contain extracts from Tg-hCBS mice [18].

correlation was low. When comparing carfilzomib vs bortezomib on these graphs, we consistently observe that carfilzomib treated mice have lower liver CBS activity for the same level of proteasome inhibition. This was somewhat surprising as bortezomib was more effective at restoring CBS activity (see discussion).

## Low dose/long-duration studies

For both the carfilzomib and bortezomib treatment experiments described above, we kept the treatment duration time short (one to two days) because both drugs caused signs of toxicity (lethargy) in the animals and longer usage resulted in morbidity and death. Therefore, we examined the effects of lower dosages given for longer periods of time.

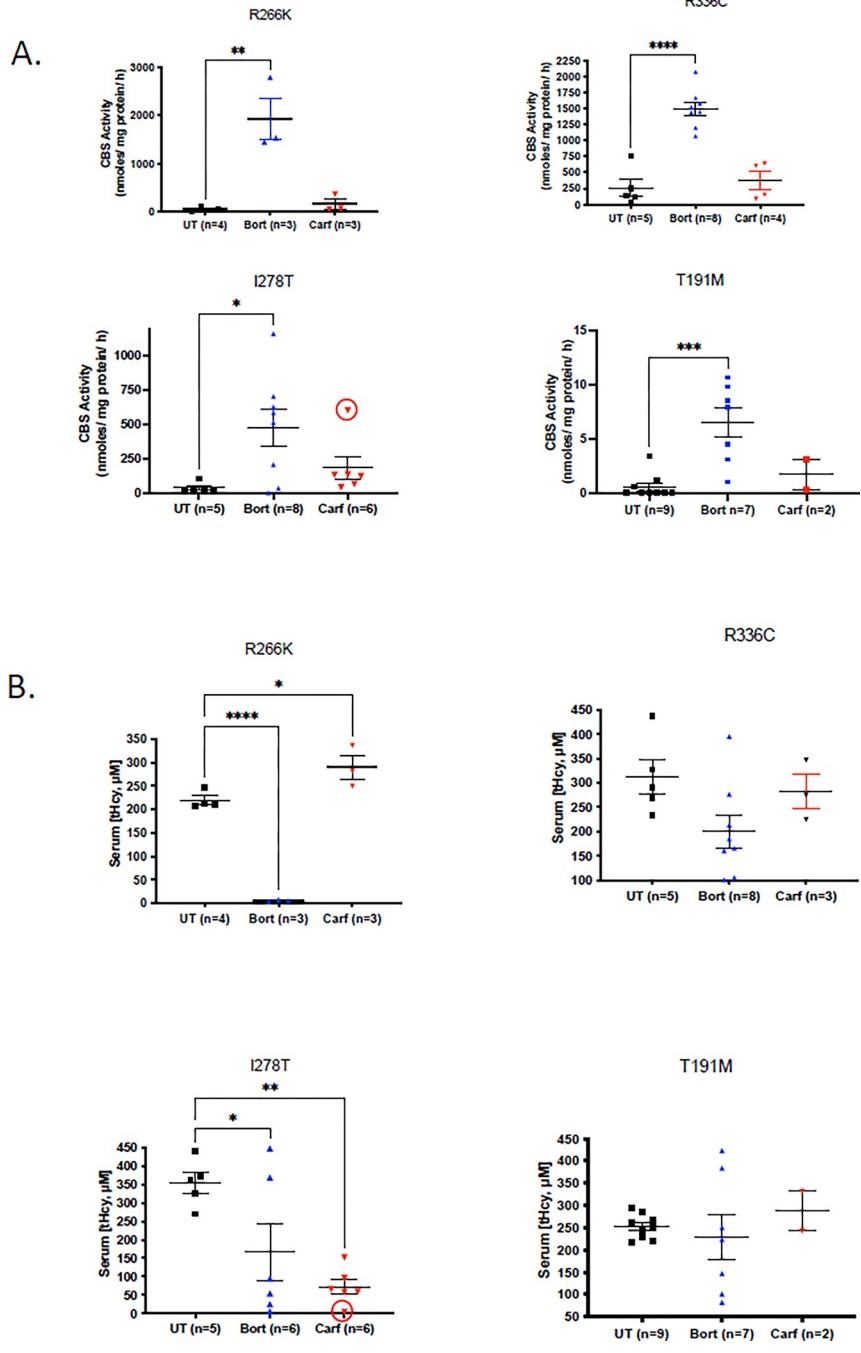

**Fig 3. Liver CBS activity and serum tHcy, in treated humanized CBS mice.** A. Liver CBS activity in indicated strains with indicated treatments. Red circle denotes highly responsive sample noted in Fig 2C. B. Serum tHcy levels in indicated strains with indicated treatments. Error bars show SEM. Asterisks indicate significant differences from untreated samples: *p<0.05, **p<0.01, ***p<0.001, and ****P<0.0001.

We first examined the effect of lower dosage bortezomib in the *Tg-I278T* mouse model. In these experiments mice were injected a total 12 times over a 32 day period with subcutaneous bortezomib initially at 0.5 mg/kg and later at 0.8 mg/kg (Fig 5A). Thus over the entire course of the study, the average total dose was 0.23 mg/kg/day, about 47% of the dose used in the short term studies. Total homocysteine was measured before the first injection (d0) and at

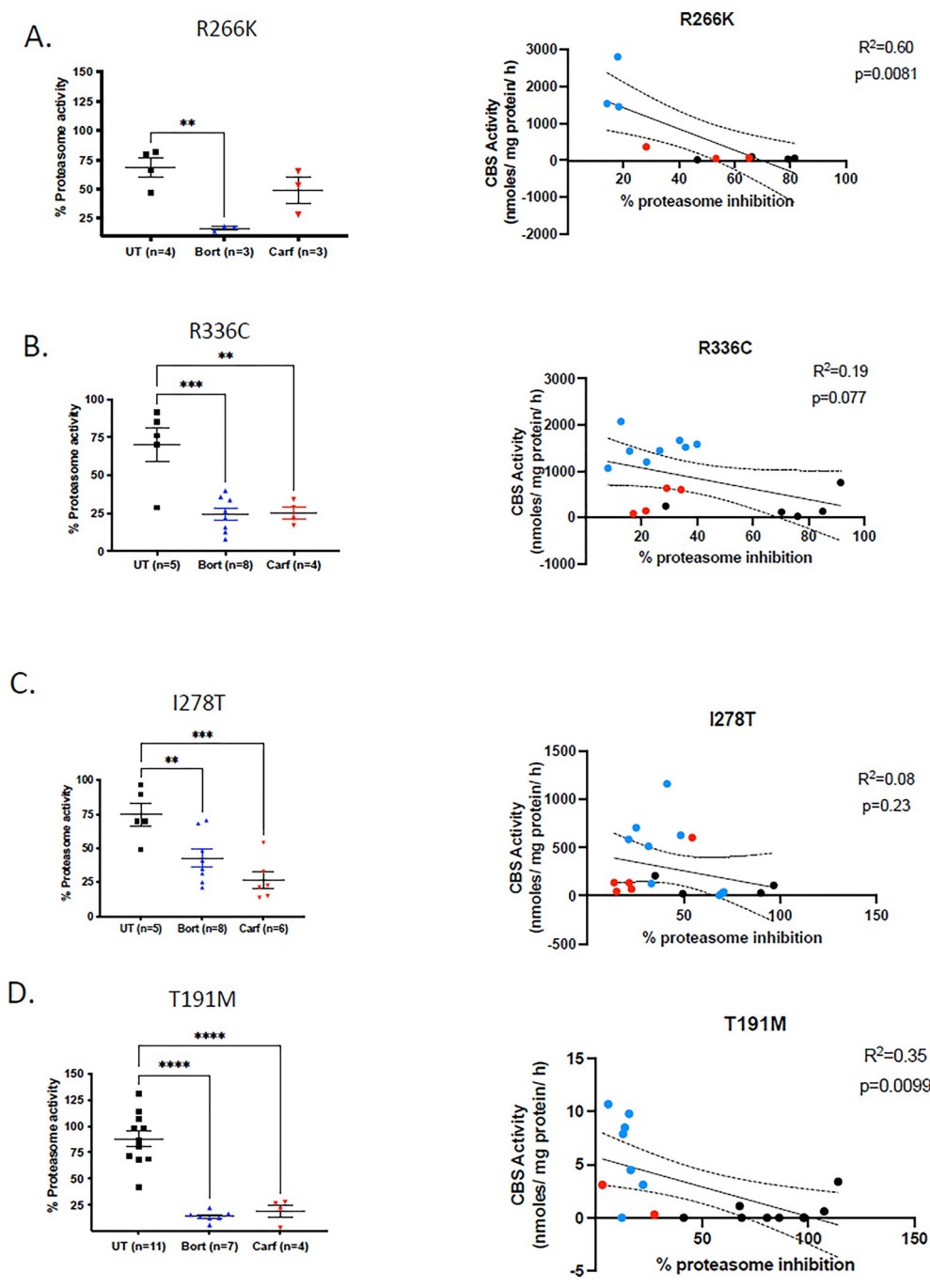

**Fig 4. Proteasome activity of bortezomib- and carfilzomib-treated mice.** A. left side shows liver proteasome activity of *Tg-R266K* liver extracts expressed as percent of untreated control *Tg-hCBS* extract. Assay was performed using a fluorescent substrate. On right side % proteasome activity is plotted on X-axis vs. liver CBS activity on Y-axis. Color coding is: Black = untreated, red = carfilzomib-treated, blue = bortezomib-treated. B. Same as above for *Tg-R336C*. C. Same as above for *Tg-I278T*. D. Same as above for *Tg-T191M*.

d11, d18, d25 and d32 (Fig 5B). Over the entire course of the experiment we lowered tHcy from a mean of 345 μM to 179 μM. The lowering showed a linear trend with time (P<0.001), suggesting that the effectiveness of the treatment increased as the total dose increased. The average weight of the animals did not change through d30, however, there may have been a

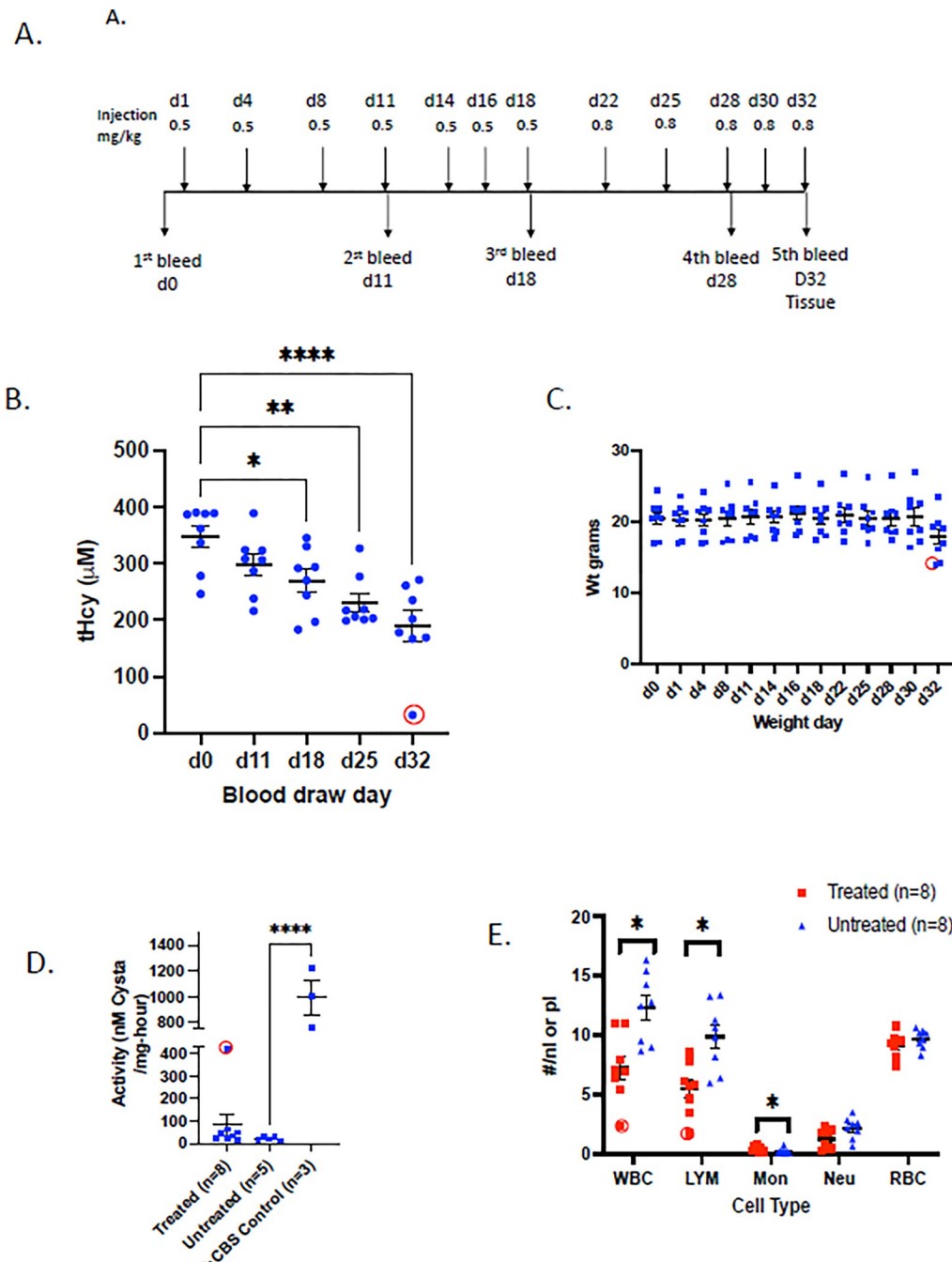

**Fig 5. Long-term, low-dose bortezomib in *Tg-I278T* mice.** A. Diagram showing injection and blood collection times. Injections amounts and times are shown above the line, blood and tissue collection is shown below. B. tHcy at indicated time. Significance was determined by as different from d0 time point. C. Weight of animals during experiment. D. CBS activity. Red circle shows points for animal that appeared moribund at time of sacrifice. E. Blood counts at end of experiment. WBC = white blood cells; LYM = lymphocytes; Mon = monocytes; Neu = neutrophils; RBC = red blood cells.

small decrease at d32 (Fig 5C). At the end of the experiment, liver CBS activity was examined relative to zinc-induced *Tg-I278T* control animals (Fig 5D). With one exception (see below), we did not observe much difference between tested and untreated animals. At the end of the

study, we also performed blood counts on the animals examining WBC, lymphocyte, monocyte, neutrphyll and RBC levels (Fig 5D). Treated animals had significantly reduced levels of WBCs and lymphocytes compared to controls, indicating that even at low dosage there is immune toxicity associated with the drug.

Interestingly, one of the eight treated animals did show CBS activity significantly outside the range determined in the untreated controls. This same animal also showed the lowest tHcy at the end of the experiment, the most weight-loss, and the lowest WBC and lymphocyte counts (see circled point in Fig 5B-5E.). This finding suggests that the strength of response to bortezomib may be linked to toxicity. Overall, our data indicates that lower dose bortezomib is better tolerated by the mice, but is also less effective at restoring function to mutant CBS.

A follow-up experiment was performed to determine if we could increase the effectiveness of low-dose bortezomib by combining it with bimoclomol. Bimoclomal is a non-toxic hydroxylamine-derivative that acts as a co-inducer of heat shock proteins, and has been shown to have beneficial effects in rodent models of vascular disease and diabetes [22]. We performed a two-arm study in which *Tg-I278T* mice were treated with either low dose bortezomib alone (0.8 mg/kg, 2x per week) or low dose bortezomib in combination with bimoclomol added to their drinking water (250 mg/L) (Fig 6A). After 14 days, we found that tHcy was lowered similarly in both groups, indicating that the addition of bimoclomol did not enhance the activity of low dose bortezomib (Fig 6B). Similar to our previous long-term study, we did not observe a statistically significant difference in CBS activity in mouse liver extracts even thought tHcy was lowered (Fig 6C). Examination of various Hsp proteins revealed that Hsp70 and Hsp27 were clearly induced by the bortezomib, but we did not observe a noticeable increase with the addition of bimoclomol (Fig 6D).

We also performed some low dose/long term studies using carfilzomib and the *Tg-S466L* mouse model. In this experiment carfilzomib was given at a dose of 2.1 mg/kg/day for one week using an subcutaneously implanted osmotic pump. No significant differences were observed in either the tHcy or CBS activity levels compared to untreated control mice (Fig 7A and 7B). Western blot analysis revealed that some of the carfilzomib-treated samples had a very slight increase in CBS protein compared to untreated samples, but the overall amount was much lower than in controls and no Hsp70 induction was observed (Fig 7C). This experiment show that low dose carfilzomib is not as effective as high dose carfilzomib in rescuing *Tg-S466L* CBS activity.

## Discussion

The overall goal of the experiments described in this report was to use humanized mouse models of CBS deficiency to gain insight into whether PI drugs might be useful in the treatment of CBS deficiency. In earlier work, our group had shown that several humanized mouse models treated with high doses of the PI bortezomib, showed significant lowering of plasma tHcy and increases in the liver CBS activity [11–13]. However, in humans, bortezomib has significant side effects that would make it a less than ideal drug to treat a life-long genetic condition such as CBS deficiency. Carfilzomib is a second-generation FDA approved proteasome inhibitor that has higher on target specificity and fewer side effects than bortezomib and thus, in theory, might be more useful [23].

We first tested carfilzomib in mice expressing the human CBS p.S466L mutant protein. From previous work, we know that this protein is highly enzymatically active in both bacteria and yeast, but in mice the protein is unstable and is rapidly degraded [15,24]. We found that Carfilzomib was very effective at rescuing this allele. *Tg-S466L* mice treated with high dose carfilzomib (10 mg/kg) had liver CBS activity as high as those found that in some in *Tg-hCBS*

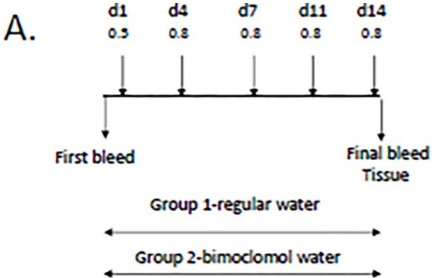

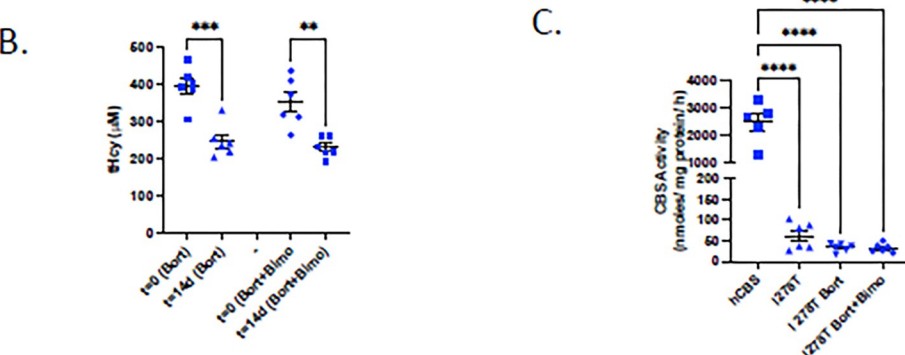

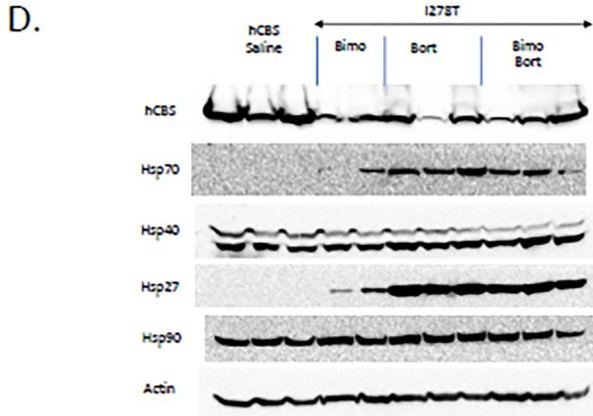

**Fig 6. Low-dose bortezomib combined with bimoclomol in *Tg-I278T* mice.** A. Treatment scheme. Injection days and amounts shown on top. B. tHcy pre and post treatment. C. Liver CBS enzyme activity at end of treatment. D. Western blot of hCBS and indicated Hsps. Note that two bimoclomol only treated animals (Bimo) are included as controls.

mice (WT hCBS) (see [25], Fig 4), and tHcy levels that were below the level of detection (<5 μM).

Although carfilzomib could restore function to p.S466L CBS in mice, we found that it was less effective than bortezomib at rescuing other mutant CBS proteins including p.I278T, p.

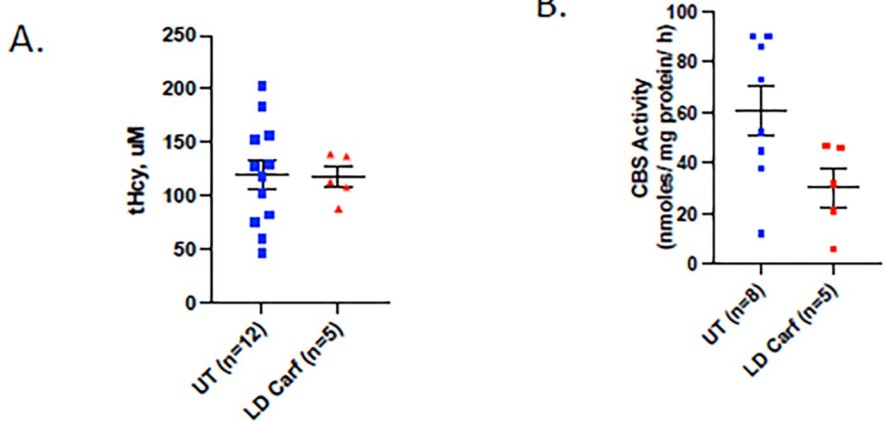

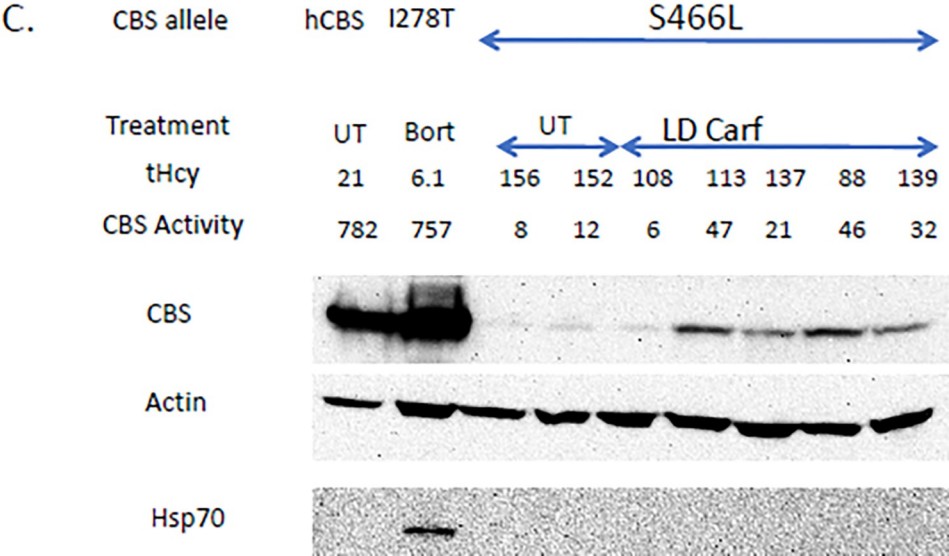

**Fig 7. Low-dose Carfilzomib in *Tg-S466L* mice.** Mice were treated at a dose of 2.1 mg/Kg/day for one week using an osmotic pump. A. Comparison of tHcy in untreated (UT) and treated (LD Carf) mice. B. Liver CBS activity. C. Western blot analysis.

R336C, p.R266K and p.T191M. The reason for this reduced effectiveness is not entirely clear, as each mutation produces a protein with different properties. Unlike p.S466L which is located in the C-terminal regulatory domain, all four of the additional mutants are located in the catalytic domain and have some impairment of enzymatic function in heterologous expression systems, as well as being unstable in mice [13,26–28]. This implies that for these alleles the protein must be both stabilized <u>and</u> refolded to be active in mouse liver. Another important

caveat in our studies is that the two drugs were administered in different doses using different methods. Both drugs were given at dosages that were similar to those used in human cancer patients when normalized for weight to surface area considerations [29]. However, bortezomib was delivered using an osmotic subcutaneously implanted pump, while carfilzomib was delivered as a i.v. bolus. The reason we chose to deliver carfilzomib as a single bolus was because we found it was a bit more effective at restoring CBS function in the initial experiments using the Tg-S466L mouse model. In an attempt to compare the pharmacologic effectiveness of the two drugs, we measured liver lysate proteasome activity. With the exception of the Tg-R266K model, liver lysates from carfilzomib-treated liver samples had similar levels of proteasome inhibition as bortezomib-treated ones. These findings suggest that the superior rescuing quality of bortezomib is due to something other than increased proteasome inhibition *in vivo*. Since bortezomib is thought to have more "off-target" effects than carfilzomib [23], it may be that these are important in stimulating proper folding of human CBS protein.

Previously, our group had shown that in *S. cerevisiae*, chaperone protein induction played an important role increasing function to mutant CBS protein. Specifically, it was demonstrated that either overexpression of the yeast Hsp70 orthologue *SSA2*, or deletion of the gene for the small yeast heat shock protein, *HSP26*, could partially rescue mutant human CBS function, even in the absence of proteasome inhibition [30]. This led to the idea that mechanism used by bortezomib to enhance mutant CBS activity might involve the induction of chaperone proteins as well as direct proteasome inhibition [21]. Therefore, we measured the steady-state protein levels of the mouse orthologs of the chaperones, Hsp70 and Hsp27. For the most part, we found that both these proteins were induced by either treatment, although there were differences in the relative levels in the different mouse models examined. For example, in *Tg-R336C* mice, carfilzomib induced much higher levels of Hsp70 and Hsp27 than bortezomib did. However, the strength of CBS rescue did not seem related to robustness of Hsp70 and Hsp27 induction. It should also be noted that we did not observe any induction Hsp40 and Hsp90 by either carfilzomib or bortezomib. These findings suggest that rescue of mutant CBS protein in mouse liver is a more complex process than in yeast, perhaps involving additional chaperone proteins that were not examined in here.

Although bortezomib, and to a lesser extent carfilzomib, treatment could increase liver CBS activity and lower tHcy in the mouse models used above, it was also observed that prolonged daily treatment with these drugs caused the mice to become lethargic and eventually morbid. Therefore, we examined the effectiveness of lower amounts of PIs in certain mouse models. We tested lower dose bortezomib using the *Tg-I278T* model and lower dose carfilzomib using the *Tg-S466L* model. For the bortezomib studies, we found that giving bortezomib twice a week by subcutaneous injection was generally well tolerated, but the rescue as judged by tHcy lowering and liver CBS activity was much weaker that high-dose bortezomib. Mean serum tHcy was lowered by about 50% in the low-dose treated animals, but was still over 170 μM (normal in mouse is 3–7 μM). We could not demonstrate that there was a significant increase in liver CBS activity, but this probably is due to the limited sensitivity of the assay at very low activity levels. However, it is possible that the tHcy lowering we observed was due to the injection and handling of the animals. We tried to increase the effectiveness of low-dose bortezomib by combining it with bimoclomol, a small molecule that has been reported to induce several heat shock proteins by interacting with the HSF 1 transcription factor [31]. However, we did not observe any difference in either rescue or Hsp induction when bimoclomol was added. Lastly, we tried lower dose carfilzomib in the *Tg-S466L* mouse model, with the same dose given over seven-day period instead of a single day. Once again, the amount of rescue was minimal, with only a slight increase in liver CBS activity and no lowering of serum tHcy. These findings suggest that in order to be effective at rescuing mutant CBS proteins,

carfilzomib must be given in high dosages similar to that used in treating cancer patients. Based on these experiments, it appears that toxicity and rescue are not separable.

Overall, the data in this paper suggests that carfilzomib is generally less effective than bortezomib in restoring function to mutant CBS allele *in vivo*. However, neither drug appears to be effective when used at non-toxic doses. Thus the idea of using these proteasome inhibitors as a monotherapy to treat CBS deficiency is probably not feasible. Future work should focus on other types of proteostasis modulators that might have less toxicity.

## Supporting information

**S1 File. Underlying data for all graphs.**
(XLSX)

**S2 File. Notebook data for Western blots.** File shows complete image data for all Western blots, as well as molecular weight migration markers.
(PDF)

## Acknowledgments

We thank Cynthia Myers for the synthesis of bimoclomol in the FCCC Organic Synthesis Facility. We thank Xiang Hua and the FCCC Transgenic Mouse Facility for production of the Tg-T191M transgenic mouse, and the FCCC Laboratory Animal Facility for mouse husbandry. We also thank Rita Michielli for mouse tail genotyping.

## Author Contributions

**Conceptualization:** Warren D. Kruger.

**Data curation:** Sapna Gupta, Hyung-Ok Lee, Liqun Wang, Warren D. Kruger.

**Formal analysis:** Sapna Gupta, Hyung-Ok Lee, Warren D. Kruger.

**Funding acquisition:** Warren D. Kruger.

**Investigation:** Sapna Gupta, Hyung-Ok Lee, Liqun Wang, Warren D. Kruger.

**Methodology:** Sapna Gupta, Hyung-Ok Lee, Liqun Wang, Warren D. Kruger.

**Project administration:** Hyung-Ok Lee, Warren D. Kruger.

**Resources:** Warren D. Kruger.

**Supervision:** Warren D. Kruger.

**Writing – original draft:** Sapna Gupta, Hyung-Ok Lee, Warren D. Kruger.

**Writing – review & editing:** Hyung-Ok Lee, Warren D. Kruger.

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
