## [Decision Letter · Decision Letter 0]

4 Apr 2023

PONE-D-23-03975Bortezomib is more effective than carfilzomib in reactivating mutant human cystathionine beta-synthase  in micePLOS ONE

Dear Dr. Kruger,

Thank you for submitting your interesting manuscript to PLOS ONE. As you will note from the following/attached reviews, your work was critiqued by two experts in the field who both saw value in the study but also raised some valid concerns. In many cases, these can be addressed textually, but I would urge you to perform the controls noted by Reviewer #2. Because Reviewer #1 was concerned by the study power, which would require additional mice/experiments, here I would suggest--as advised by both reviewers--that you soften your conclusions as needed. A further discussion of key points is also needed in places, as is additional information on the study design/experimental procedures. Should you choose to resubmit your work, please note that the revised study would be re-reviewed by one or possibly both original reviewers, so please carefully address how the text and data have been revised. In addition, please submit your revised manuscript by May 19 2023 11:59PM. If you will need more time than this to complete your revisions, please reply to this message or contact the journal office at plosone@plos.org. Please include the following items when submitting your revised manuscript:A rebuttal letter that responds to each point raised by the academic editor and reviewer(s). You should upload this letter as a separate file labeled 'Response to Reviewers'.A marked-up copy of your manuscript that highlights changes made to the original version. You should upload this as a separate file labeled 'Revised Manuscript with Track Changes'.An unmarked version of your revised paper without tracked changes. You should upload this as a separate file labeled 'Manuscript'.

Thank you again, and we look forward to receiving your revised manuscript.

Sincerely yours,

Jeffrey L Brodsky

Academic Editor

PLOS ONE

Journal Requirements:

2. Please make sure that all information entered in the 'Ethics Statement' section regarding ethics approval is also included in the Methods section of the manuscript.

"This work was funded by grants from the National Institutes of Health (DK101404 to WK and CA006927 to Fox Chase Cancer Center)." 

"This work was funded by grants from the National Institutes of Health (DK101404 and CA006927).  We thank Cynthia Myers for the synthesis of bimoclomol in the FCCC Organic Synthesis Facility.  We thank Xiang Hua and the FCCC Transgenic Mouse Facility for production of the Tg-T191M transgenic mouse, and the FCCC Laboratory Animal Facility for mouse husbandry.  We also thank Rita Michielli for mouse tail genotyping. "

"This work was funded by grants from the National Institutes of Health (DK101404 to WK and CA006927 to Fox Chase Cancer Center)." 

6. Please amend your list of authors on the manuscript to ensure that each author is linked to an affiliation. Authors’ affiliations should reflect the institution where the work was done (if authors moved subsequently, you can also list the new affiliation stating “current affiliation:….” as necessary).

7. We note that you have included the phrase “data not shown” in your manuscript. Unfortunately, this does not meet our data sharing requirements. PLOS does not permit references to inaccessible data. We require that authors provide all relevant data within the paper, Supporting Information files, or in an acceptable, public repository. Please add a citation to support this phrase or upload the data that corresponds with these findings to a stable repository (such as Figshare or Dryad) and provide and URLs, DOIs, or accession numbers that may be used to access these data. Or, if the data are not a core part of the research being presented in your study, we ask that you remove the phrase that refers to these data.

8. Please include your full ethics statement in the ‘Methods’ section of your manuscript file. In your statement, please include the full name of the IRB or ethics committee who approved or waived your study, as well as whether or not you obtained informed written or verbal consent. If consent was waived for your study, please include this information in your statement as well. 

9. PLOS ONE now requires that authors provide the original uncropped and unadjusted images underlying all blot or gel results reported in a submission’s figures or Supporting Information files. This policy and the journal’s other requirements for blot/gel reporting and figure preparation are described in detail at https://journals.plos.org/plosone/s/figures#loc-blot-and-gel-reporting-requirements and https://journals.plos.org/plosone/s/figures#loc-preparing-figures-from-image-files. When you submit your revised manuscript, please ensure that your figures adhere fully to these guidelines and provide the original underlying images for all blot or gel data reported in your submission. See the following link for instructions on providing the original image data: https://journals.plos.org/plosone/s/figures#loc-original-images-for-blots-and-gels. 

Reviewers' comments:

Reviewer's Responses to Questions

**Comments to the Author**

1. Is the manuscript technically sound, and do the data support the conclusions?

Reviewer #1: Yes

Reviewer #2: Partly

2. Has the statistical analysis been performed appropriately and rigorously? 

Reviewer #1: Yes

Reviewer #2: Yes

3. Have the authors made all data underlying the findings in their manuscript fully available?

Reviewer #1: Yes

Reviewer #2: Yes

4. Is the manuscript presented in an intelligible fashion and written in standard English?

Reviewer #1: Yes

Reviewer #2: Yes

5. Review Comments to the Author

Reviewer #1: This is a well-written manuscript and the underlying work advances our understanding of the therapeutic potential of proteosome inhibitors. However, the authors should temper their conclusions with regard to the relative efficacy of the proteosome inhibitors, which at times oversimplify and overstate the findings. There are several specific areas that should be addressed. Please see attached:

Reviewer #2: In this paper, the authors continue their long-standing work treating transgenic mice with proteasome inhibitors to examine therapeutic potential. They have previously shown for (I believe) all of the mutants except for the T191M that bortezomib (BZ) rescues HCy clearance and CBS expression and activity.

The new contribution of this paper is comparing the effects of carfilzomib (CZ) to BZ. They find that CZ impairs hepatic proteasome activity but is largely ineffective at rescuing serum HCy and restoring CBS expression and activity with the exception of the S466L and I278T mutations (more on this later). The authors also perform some low doze BZ and CZ experiments, which are not very enlightening but probably merit inclusion nonetheless.

Largely as a description of comparative results I think the paper comes to appropriate conclusions, although I think the authors need to be even more explicitly cautious in their discussion. Not only does the dosing and route of administration potentially affect their outcomes, but so too does the fact that they are selecting only one time point for examination, and I could easily believe that the pharmacokinetics of the drug, combined with the synthesis and turnover of CBS, might yield different results if a more systematic examination of time points were conducted. The larger point of the paper, though, is that, despite the fact that it seems to inhibit the proteasome fairly robustly, CZ is not at least in these assays nearly as protective as BZ, which implies at least at face value that BZ works through proteasome-independent mechanisms. The authors mention this in the discussion, but I am a bit surprised that conclusion is not more front-and-center in the abstract.

Points for attention:

1. For CZ-treated animals were control mice injected and/or pumped (with water)? The methods do not indicate this, though they explicitly state that the BZ controls were pumped with diluent. If the controls for CZ were not so-treated, that would be in my opinion a pretty serious problem of rigor.

2. Related to the above, the experiment in Fig. 5 lacks (at least for the serum HCy and body weight) the control of animals injected with diluent on the same schedule. It is hard to interpret especially 5B without such a control. I can imagine that the stress of being repeatedly handled and jabbed might affect any number of physiological parameters. If that control wasn’t done, it either needs to be done or the experiment should not be included.

3. Figure 6—it isn’t clear that the Bimo had any effects on the system at all. A simple explanation is that the compound is not active as intended. Absent a way to confirm that it is doing something in vivo, it’s a really difficult experiment to interpret and I would take it out.

4. The I278T animals are strange. CZ diminishes serum HCy but without any increase in CBS expression or activity. How do the authors account for this?

5. Gels need migration standards.

6. Label font sizes in Fig. 2 are discrepant and the correlation graphs in Fig. 4 look like they were dropped straight from Excel. Showing treated animals to the left in 5E is a bit unconventional, and I would recommend keeping color designations (UT, BZ, CZ) consistent in Fig. 3. Overall for aesthetics I think the figures should be cleaned up.

6. PLOS authors have the option to publish the peer review history of their article (what does this mean?). If published, this will include your full peer review and any attached files.

Reviewer #1: No

Reviewer #2: No

---

## [Editor Report · Decision Letter 1]

25 Apr 2023

PONE-D-23-03975R1Examination of two different proteasome inhibitors in reactivating mutant human cystathionine beta-synthase in micePLOS ONE

Dear Dr. Kruger,

Thank you for re-submitting your manuscript to PLOS ONE. I have read carefully over the manuscript in light of the reviewers' comments, and prior to acceptance would suggest that the following minor issues are addressed. 1. Please edit/clarify the following added text, "in inducing in inducing", and "that the lowering..."2. Please add some additional details to the construction of the mouse strain in the Methods, with regard to Reviewer 1's comment on the background.3. Please be specific in the text with regard to "almost all the mice"...numbers would be helpful here.4. Based on the suggested title change, please add a sentence or two placing your work in the context of a prior paper on the effects of different proteasome inhibitors in another conformational disease (PMID: 18775310) Please submit your revised manuscript by Jun 09 2023 11:59PM. If you will need more time than this to complete your revisions, please reply to this message or contact the journal office at plosone@plos.org. Please include the following items when submitting your revised manuscript:A rebuttal letter that responds to each point raised by the academic editor and reviewer(s). You should upload this letter as a separate file labeled 'Response to Reviewers'.A marked-up copy of your manuscript that highlights changes made to the original version. You should upload this as a separate file labeled 'Revised Manuscript with Track Changes'.An unmarked version of your revised paper without tracked changes. You should upload this as a separate file labeled 'Manuscript'.If applicable, we recommend that you deposit your laboratory protocols in protocols.io to enhance the reproducibility of your results. Protocols.io assigns your protocol its own identifier (DOI) so that it can be cited independently in the future. For instructions see: https://journals.plos.org/plosone/s/submission-guidelines#loc-laboratory-protocols. Additionally, PLOS ONE offers an option for publishing peer-reviewed Lab Protocol articles, which describe protocols hosted on protocols.io. Read more information on sharing protocols at https://plos.org/protocols?utm_medium=editorial-email&utm_source=authorletters&utm_campaign=protocols.

Thank you again for submitting this interesting study to PLOS ONE, and I trust that these final changes will not be too laborious.

Sincerely yours,

Jeffrey L Brodsky

Academic Editor

PLOS ONE
---

## [Author Response · Author response to Decision Letter 1]

2 May 2023

I have made all four of the corrections listed in the decision letter.

---

## [Editor Report · Decision Letter 2]

18 May 2023

Examination of two different proteasome inhibitors in reactivating mutant human cystathionine β-synthase in mice

PONE-D-23-03975R2

Dear Dr. Kruger,

Thank you for making these final edits to your paper--we are delighted to accept the paper for publication.

Within one week, you’ll receive an e-mail from the journal detailing any required amendments. When these have been addressed, you’ll receive a formal acceptance letter and your manuscript will be scheduled for publication.

Kind regards,

Jeffrey L Brodsky

Academic Editor

PLOS ONE

---

## [Editor Report · Acceptance letter]

6 Jun 2023

PONE-D-23-03975R2 

Examination of two different proteasome inhibitors in reactivating mutant human cystathionine β-synthase in mice 

Dear Dr. Kruger:

I'm pleased to inform you that your manuscript has been deemed suitable for publication in PLOS ONE. Congratulations! Your manuscript is now with our production department. 

Kind regards, 

on behalf of

Dr. Jeffrey L Brodsky 

Academic Editor

PLOS ONE